# Management of Skin Lesions in Patients with Epidermolysis Bullosa by Topical Treatment: Systematic Review and Meta-Analysis

**DOI:** 10.3390/healthcare12020261

**Published:** 2024-01-19

**Authors:** Manuel Pabón-Carrasco, Rocio Caceres-Matos, Marta Roche-Campos, Maria Antonia Hurtado-Guapo, Mercedes Ortiz-Romero, Luis M. Gordillo-Fernández, Daniel Pabón-Carrasco, Aurora Castro-Méndez

**Affiliations:** 1Research Group PAIDI-CTS-1054: “Interventions and Health Care, Red Cross (ICSCRE)”, Nursing Department, Faculty of Nursing, Physiotherapy and Podiatry, University of Seville, 6 Avenzoar ST, 41009 Seville, Spain; mpabon2@us.es; 2Research Group PAIDI-CTS-1050: “Complex Care, Chronicity and Health Outcomes”, Nursing Department, Faculty of Nursing, Physiotherapy and Podiatry, University of Seville, 6 Avenzoar ST, 41009 Seville, Spain; 3Hospital Universitario de Cruce, 48903 Bilbao, Spain; marta11roche@gmail.com; 4Nursing Department, Faculty of Nursing, Extremadura University, 06006 Badajoz, Spain; ahurtado@unex.es; 5Faculty of Nursing, Physiotherapy and Podiatry, University of Seville, 41009 Seville, Spain; mortiz17@us.es (M.O.-R.); lgordillo@us.es (L.M.G.-F.); auroracastro@us.es (A.C.-M.); 6Crux Roxa Rehabilitación, 41008 Sevilla, Spain; danipabon94@gmail.com

**Keywords:** epidermolysis bullosa, lesions bullous, topical administration, review systematic, meta-analysis

## Abstract

Epidermolysis bullosa (EB) is the overarching term for a set of rare inherited skin fragility disorders that result from mutations in at least 20 different genes. Currently, there is no cure for any of the EB subtypes associated with various mutations. Existing therapies primarily focus on alleviating pain and promoting early wound healing to prevent potential complications. Consequently, there is an urgent need for innovative therapeutic approaches. The objective of this research was to assess the efficacy of various topical treatments in patients with EB with the goal of achieving wound healing. A secondary objective was to analyse the efficacy of topical treatments for symptom reduction. A literature search was conducted using scientific databases, including The Cochrane Library, Medline (Pubmed), Web of Science, CINHAL, Embase, and Scopus. The protocol review was registered in PROSPERO (ID: 418790), and inclusion and exclusion criteria were applied, resulting in the selection of 23 articles. Enhanced healing times were observed compared with the control group. No conclusive data have been observed on pain management, infection, pruritus episodes, and cure rates over time. Additionally, evidence indicates significant progress in gene therapies (B-VEC), as well as cell and protein therapies. The dressing group, Oleogel S-10, allantoin and diacerein 1%, were the most represented, followed by fibroblast utilisation. In addition, emerging treatments that improve the patient’s innate immunity, such as calcipotriol, are gaining attention. However, more trials are needed to reduce the prevalence of blistering and improve the quality of life of individuals with epidermolysis bullosa.

## 1. Introduction

Epidermolysis bullosa (EB), or butterfly skin, is a chronic and incurable hereditary disease caused by an inherited genetic mutation, with a prevalence ratio of 1 individual per 50,000 people, making it classified as a rare disease [1,2,3,4]. Its systemic clinical manifestations are diverse, leading to nutritional compromise due to gastrointestinal disorders, oral and dental disorders, and abnormalities in the internal epithelial lining in organs [5]. In severe cases, patients may require a nasogastric tube or gastrostomy due to a significant protein deficit, which, in turn, delays wound healing and causes skin dryness. The individual’s ability to heal may be compromised by malnutrition, anaemia, pruritus, and pain, and must be appropriately treated [6,7,8].

As a genodermatosis, it induces fragility, resulting in mucocutaneous blisters, erosions, and ulcers in response to mild trauma or friction. This fragility gives rise to the designation of “butterfly skin” [1,3]. Dermal symptoms are varied, including blisters, ulcersand erosions that easily complicate with any skin trauma, friction, or dryness. The impact of EB can limit the patient’s life due to pain from the lesions, alterations in body image, and resulting restrictions, significantly affecting health, quality of life, and psychosocial well-being for the individual and his surroundings [9,10].

Various hereditary genetic mutations that cause EB have been identified, classifying the disease into four main types (involving up to 20 different genes). Precise diagnosis depends on correlating clinical, microscopic, electron, and immunohistological characteristics with mutational analysis. EB is classified into four subtypes that determine the location, severity, and prognosis of the lesions, which can lead to the development of chronic wounds that may progress to squamous cell carcinomas [5]. More than 30 subtypes are recognised, grouped into four main categories based predominantly on the cleavage plane within the skin, reflecting the underlying molecular anomaly: Simple EB, Junctional EB, Dystrophic EB, and Kindler EB. The simple type affects the most superficial layers, the junctional type (Type 2) is located in clefts in the lamina lucida of the skin, the dystrophic type (Type 3) manifests with blisters and can affect nails and skin, and finally, Kindler syndrome is characterised by keratoderma and, at times, is associated with mental retardation; in severe cases, EB can lead to early death [9,11,12] (Section A.1.).

Given its chronic and incurable nature, treatment is palliative from a dermatological point of view, the treatment of painful lesions is considered a challenge due to infections and slow healing. In addition, this may be influenced by nutritional deficiencies that hinder wound healing. Current clinical practice focusses on the control of symptoms, signs, and complication through topical treatment. Wound care, fragile skin protection, pain and itch management, dryness, infection control, and trauma prevention are crucial [13,14,15].

The care of these patients requires specific attention to the lesions of the disease, which is a challenge in daily clinical activity. This situation is exacerbated by the lack of consensus on the approach to local lesions [14]. Similarly, the choice of wound management strategies must balance efficacy, patient choice, and quality of life in a cost-effective way [16,17]. The current care protocol consists of bandaging and cleaning open wounds to prevent infection, pain management, and symptomatic treatment of complications [18,19].

In fact, the current local therapeutic is controversial and complex, mainly focused on a bandage application to protect the lesions and symptoms controls (especially pain and itching) and the application of diacerein [20], Oleogel S-10 [21], allantoin (e.g., product SD-101) [22], calcipotriol [23], Beremagene geperpavec (B-VEC) [24], silicone dressings or hydrogel dressings or with dialkylcarbamoyl chloride (DACC) [6,25] or henna [26], among others [27,28,29]. Considering the above, this work is considered relevant because the topical treatments used are experimental, as there is no consensus on their effectiveness, which allows the establishment of a standardised action protocol [5,15].

All the above highlights the need to provide current scientific evidence on new treatments used in wound healing, allowing recommendations for clinical practice. Therefore, this systematic review with meta-analysis aims to evaluate the effectiveness of local treatments on wounds in patients with EB.

## 2. Methods

### 2.1. Search Strategy and Inclusion Criteria

Two studies (M.A.H-G, M.R-C) independently conducted a comprehensive search of databases: Medline (PubMed), CINAHL, EMBASE, Web of Science, and The Cochrane Library, following the recommended guidelines in Preferred Reporting Items for Systematic Reviews and Meta-Analyses for Protocols (PRISMA) [30]. Furthermore, ClinicalTrials.gov was reviewed to assess the existence of ongoing and published clinical trials, as well as their registration in the PROSPERO database (CRD42023418790). Google Scholar was also examined to identify unconventional literature and mitigate potential publication biases.

These search terms were obtained from the Medical Subject Headings thesaurus (MeSH) as: (Epidermolysis Bullosa OR Butterfly skin) AND (Skin Care OR Wound Healing OR wound dressings OR “Oleogel-S10” OR Ointments OR Calcipotriol OR diacerein OR allantoin OR skin cream OR gentamicins OR genetic therapy OR fibroblasts OR protein therapy OR administration, topical) and their equivalent in Spanish or French. Searches were conducted from November to December 2023 by two researchers. The inclusion criteria comprised articles published without year restriction, in English or Spanish, related to the objectives of the study, and randomised clinical trials (RCTs) conducted in humans.

The selective inclusion of RCTs was executed to elevate the methodological precision of the review and alleviate biases in an inherently intricate subject. This analysis was complex in relation to various factors that influence its results, including aspects such as the time of complete wound healing; changes in the Wong–Baker FACES scores (WBF) for pain assessment; the proportion of wound healing within a specified timeframe; changes in the Body Surface Area scores (BSA) and infections and cases of pruritus. Body surface area (BSA) is calculated using a formula that provides an estimate of body surface area in square meters based on a person’s height and weight. On the other hand, Wong–Baker FACES patients are asked to choose the face that best describes their pain level at that moment. The scale generally consists of six faces, each with a different facial expression. It is often accompanied by an interval of verbal intensity, which can range from “no pain” to “maximum intensity of pain”.

### 2.2. Data Extraction

Ethical approval was unnecessary for this study, as it constitutes a systematic review with meta-analysis and does not involve direct patient participation. The search and selection of articles were carried out independently by two researchers, and any disagreements were resolved through consultation with an expert in wound cicatrisation and cutaneous integrity injuries (M.P-C). The initial screening involved a review of titles and abstracts, followed by a comprehensive evaluation of the full articles. Furthermore, a bibliographic search was conducted both forward and backward within the references cited in the chosen studies. The level of agreement between the two researchers in assessing the appropriateness of the studies was quantified using the Kappa statistical test.

A data coding manual was followed to collect information from each study, including (1) author’s name; (2) year of publication; (3) country of origin; (4) study design; (5) sample size; (6) type of intervention (use of local treatment versos control group using placebo); (7) participants’ age; (8) objectives of each study; and (9) outcomes obtained. The primary continuous outcomes analysed included results focused on the time of complete wound healing; changes in WBF scores for pain assessment; the proportion of wound healing within a specified timeframe; changes in BSA scores; and dichotomous analyses included infections and pruritus outcomes.

### 2.3. Quality and Bias Risk Assessment

The Cochrane Risk of Bias Tool [31] was used, categorising the level of risks into three levels: low, high, or unclear. These risks included random sequence generation, allocation concealment, blinding of participants and personnel, blinding in outcome assessment, integrity of outcome data, selective reporting, and other potential sources of bias. Studies without a high risk of bias in any category were classified as high quality (1++), those with a high risk or two unclear risks as medium quality (1+), and others as low quality (1−).

For bias assessment, the Cochrane Handbook for Intervention Reviews (Revman^®^ Version 5.4) was used. Two independent reviewers subjectively evaluated articles, assigning ratings of “high,” “low” or “unclear” based on selection, performance, detection, attrition biases, and other potential biases. Discrepancies were resolved through discussions, reaching consensus. If consensus was not achieved, the opinion of a third investigator was sought.

Statistical analysis and bias assessment were performed using Review Manager software, version 5.4^®^ (Cochrane Library, London, UK). Furthermore, data were imported into the Grade Pro^®^ application (https://www.gradepro.org/) to evaluate the recommendation grade [32].

### 2.4. Data Synthesis and Statistical Analysis

The Odds Ratio (OR) was employed for the comparison of dichotomous variables, and 95% confidence intervals (CI) were provided. Continuous variables were assessed using mean differences (MD) along with a 95% CI. In cases where standard deviation data were unavailable in the study, the method recommended by Hozo et al. [33] was applied.

Both binary and continuous data were computed using fixed or random-effects models. The fixed-effect model was initially selected in the absence of significant heterogeneity between studies (I^2^ ≤ 50%). Alternatively, the random-effects model was applied when significant heterogeneity was present [34].

The heterogeneity between the studies was evaluated using chi-square tests and the I^2^ test, with a level of statistical significance of *p*-value < 0.05. I^2^ values ranging from 0% to 25% indicated low heterogeneity, 25% to 75% moderate heterogeneity and more than 75% high heterogeneity [35].

The results of the meta-analysis were presented using a forest plot and a funnel plot was used to assess potential publication bias between studies. The asymmetry of the funnel plot was analysed using the representation of the funnel plot and assessed with Egger’s test, considering a statistical significance level of *p*-value < 0.05 as indicative of evidence of publication bias.

The subgroup analysis based on the evolution of the wounds and side events was. Additionally, a sensitivity analysis was performed to assess the robustness of the results by sequentially omitting each study. *p*-values <0.05 were considered statistically significant.

Data for dichotomous outcomes were aggregated using a random-effects model to provide a more conservative estimate of the effects of local treatments in EB. Comparison of the impact of local treatments compared with the placebo group was expressed as the OR of infections and pruritus outcomes (number of cases) and mean with 95% confidence intervals for the time of complete wound healing (days); changes in WBF scores for pain assessment; the proportion of wound healing within a specified time-frame (%) and changes in BSA scores.

## 3. Results

### 3.1. Results Obtained in the Selection of Articles

In the initial literature search, a total of 3354 articles were identified, with three additional documents included from specific clinical trial registries (Clinical Trial Gov). After removing 155 duplicate articles using the Zotero 6.0.30^®^ reference manager, applying inclusion criteria and evaluating the titles and abstracts of the articles, 3177 were excluded for not meeting the inclusion criteria. Finally, 24 studies were selected for the systematic review analysis, of which 10 provided data for the meta-analysis. The flow diagram (Figure 1) illustrates the review process. There was excellent agreement between researchers about the eligibility assessment of the trials (Kappa statistic = 0.94).

### 3.2. Descriptive Analysis of the Results Found

The years with the highest scientific production were 2019, with six articles and 2022 and 2021, with four and three publications, respectively. The levels of evidence assessed based on the quality of the selected articles received a score of 1++ in 66.7% (n = 15) of cases, 25.0% received a score of 1+ (n = 6), and 8.3% received a score of 1− (n = 2).

The included studies addressed the time of complete wound healing (n = 5), the change in the WBF score for pain assessment (n = 3), the proportion of wound healing within a specified timeframe (n = 6), changes in the BSA scores (n = 4), and treatment in infections and pruritus (n = 4). The details of each item included are provided in Table 1.

A very low degree of recommendation was observed for overall outcomes assessed. Variables regarding the degree of recommendation and the analyses were performed using Grade PRO^®^ (Table 2).

In the intervention group, the risk (and its 95% confidence interval) is based on a comparison between the relative effect on the intervention and the risk assumed in the comparison group. A very low grade of recommendation was identified in all the outcomes evaluated, according to GRADE (Table 2).

### 3.3. Bias Risk Assessment of the Selected Studies and Publication Bias

The risk of bias was assessed using RevMan 5.4^®^, represented in Figure 2 and Figure 3 by bias assessment plots of all included studies and by a one-to-one summary plot. Allocation concealment was evident in approximately 60% of the included studies, with approximately 70% blinding of participants and staff, and 70% blinding of outcome evaluation. In relation to publication bias, a funnel plot for each study objective assessed shows an inverted funnel, with the strongest studies concentrated in the centre (Section A.2.).

### 3.4. Results of the Meta-Analysis

#### 3.4.1. Efficacy of Topical Treatments in the Time of Complete Wound Healing

In five clinical trials involving 458 wounds, with 228 in the intervention group and 230 in the control group, the efficacy of utilizing topical treatments, as opposed to the use of a placebo, in wounds in individuals with EB was assessed. Two studies exhibited a low risk of bias, while the remaining three showed a moderate level of risk [38,39,44,47,50].

A shorter healing time was observed in the group of wounds treated with topical treatments in all included studies, compared with the control group treated with a placebo.

Statistically significant differences were found between the two groups. A MD of −6.19 was obtained, with a 95% confidence interval of −8.87 to −3.51 (*p* < 0.001), and significant heterogeneity among the studies (I^2^ = 52%, *p* < 0.07) (Figure 4). This seems to indicate that existing topical treatments tend to reduce the healing time of wounds in individuals with EB.

#### 3.4.2. Efficacy of Topical Treatments on the Change in WBF Score for Pain Assessment

In three clinical trials that involved 378 wounds, with 186 in the intervention group and 192 in the control group, the effectiveness of using topical treatments, as opposed to the use of a placebo, was evaluated on the change in the WBF score for the assessment of pain in individuals with EB was evaluated. Two studies demonstrated a low risk of bias, while the remaining one exhibited a moderate level of risk [47,50,54].

On day 7 after the initiation of topical treatment, a study showed a slight trend of better WBF scores for pain assessment in the experimental group, while another study had a clear bias toward the control group that employed a placebo. However, statistically significant differences were found between the two groups, with the diamond favouring the control. An MD of 0.30 was obtained, with a 95% confidence interval of 0.20 to 0.40 (*p* < 0.001), without heterogeneity between the studies (I^2^ = 0%, *p* < 0.61).

At 14, 30, 60, and 90 days, only one study assessed the change in WBF scores for pain assessment in the group that used topical treatment compared with the placebo group, showing better scores for the intervention group. At 45 days, only one study was included, which did not show a statistically significant association, since the diamond touched the no-effect line.

Ultimately, no statistically significant differences were observed between the experimental and control group (*p* < 0.09). This is evident in the confidence intervals and in the forest plot (no-effect line) (Figure 5). These results point to a lack of consensus between the two included studies regarding the effectiveness in reducing pain when treating EB wounds with topical treatments.

#### 3.4.3. Efficacy of Topical Treatments on the Proportion of Wound Healing within a Specified Timeframe

In six clinical trials involving 538 participants, 265 in the intervention group and 273 in the control group, the efficacy of topical treatments was evaluated in the proportion of healed wounds within a specified timeframe. Five studies showed a low risk of bias [43,44,47,54], while the remaining study exhibited a moderate risk of bias [51].

At 30 days, three studies demonstrated a better progression of wound healing in the intervention group, while one did so in the placebo group. Statistically significant differences were found between the two groups, with an MD of 0.66 obtained, a 95% confidence interval of 0.44 to 1.00 (*p* = 0.05), and no heterogeneity between studies (I^2^ = 0%, *p* = 0.87).

At 60 days, all the confidence intervals of the included studies touch the no-effect line; however, four of them show a trend in favour of the intervention group. An MD of 0.75 was obtained, with a 95% confidence interval of 0.52 to 1.07 (*p* = 0.12), and there was no heterogeneity between the studies (I^2^ = 0%, *p* = 0.75). A MD of 0.75 was obtained, with a 95% confidence interval of 0.52 to 1.07 (*p* = 0.12), and no heterogeneity between studies (I^2^ = 0%, *p* = 0.75).

Regarding wound progression at 90 days, three studies showed a slight trend to the intervention group, while three did so in the placebo control group, specifically the study by Guide et al. A MD of 1.19, with a 95% confidence interval of 0.85 to 1.66 (*p* = 0.32), was obtained, and heterogeneity between studies (I^2^ = 69%, *p* < 0.05).

Ultimately, no statistically significant differences were observed between the experimental and control groups (*p* = 0.19). This is evident in the confidence intervals as well as in the forest plot (no-effect line) (Figure 6). These results suggest a lack of consensus on whether existing topical treatments are more effective in the proportion of healed wounds within a specific time period compared with the control group that used a placebo treatment.

#### 3.4.4. Efficacy of Topical Treatments in Changes in BSA Scores

In four clinical trials with 660 wounds, with 320 in the intervention group and 340 in the control group, the efficacy of using topical treatments, rather than the use of a placebo, was evaluated for changes in BSA scores. All studies had a low risk of bias [5,47,50,54].

At 30 and 60 days, the difference in BSA scores was measured only in one study at both time points. At 30 days, the BSA scores did not show significant differences between the experimental and control groups, while at 60 days, these differences were greater in the study that employed local treatment compared with placebo.

At 90 days, in four of the comparisons, the differences in BSA scores were greater in the experimental group than in the control group, while in one study, they were greater for the group that used placebo.

Finally, a MD of −0.70 was obtained, with a 95% confidence interval of −1.71 to 0.31 (*p* < 0.17), and significant heterogeneity between the studies (I^2^ = 92%, *p* < 0.001) (Figure 7). These results suggest that there is not enough evidence to determine whether topical treatments are effective in increasing BSA scores within a specified time when compared with the control group.

#### 3.4.5. Efficacy of Topical Treatments in Infections and Pruritus

Four clinical trials, with a total of 461 participants, were conducted to assess the efficacy of topical treatments compared with a placebo in the management of infections (n = 89) and pruritus (n = 30). The intervention group consisted of 227 participants, while the control group comprised 234 participants. All four studies demonstrated a low risk of bias [47,51,55,58].

Regarding the incidence of infections, one study reported a higher number of cases in the group that received local treatments, while another study showed a higher incidence in the placebo group. The remaining two studies did not show a statistically significant association, as indicated by the diamond touching the line without effect. The overall OR was calculated to be 0.96 (95% CI, 0.59 to 1.56), without heterogeneity between studies (I^2^ = 0%, *p* < 0.40). Consequently, the results did not indicate a clear preference for local treatment or placebo.

In the subgroup analysis of pruritus, all studies demonstrated a higher number of cases of pruritus in the placebo group. However, the overall results were inconclusive, approaching the threshold of no effect. The OR for pruritus was 1.56 (95% CI, 0.74 to 3.29), with no heterogeneity between studies (I^2^ = 0%, *p* < 0.90), suggesting that there is no consistent trend that favours either group.

Finally, no statistically significant differences were observed favouring local treatments over placebo were observed (*p* = 0.61, OR = 1.11 (95% CI, 0.74 to 1.67)) (Figure 8). These results suggest a lack of consensus regarding the effectiveness of the evaluated topical treatments for preventing EB wound infection compared with the control group. Regarding itching, there appears to be a trend indicating that topical treatments may be more effective in preventing the onset of itching.

## 4. Discussion

The results show statistically significant differences in healing time when comparing topical treatments with the use of saline or ointments without active ingredients. Particularly, all applied treatments demonstrate a decrease in healing time, whether using dressings (such as biocellulose dressings or DACC-coated cotton acetate dressing), gene therapy (B-VEC), or substances such as birch bark extract (Oleogel-S10) or SD-101 (6% allantoin). However, conclusive data on pain management, infection, episodes of pruritus, and cure rates over time are not available. Among advanced therapies, there is a major trend to investigate Oleogel-S10 (number of studies included in this review = 3), Diacerin (number of studies included in this review = 3) and allantoin (number of studies included in this review = 3).

Emerging trials include the use of calcipotriol (n = 1) to enhance innate immunity, gene therapy through B-VEC (n = 2), and the application of fibroblasts (n = 2). Furthermore, non-adherent dressings promoting better healing and pain reduction (n = 5) continue to be advocated.

In conclusion, we identified isolated trials exploring the use of henna, tissue engineering grafts, and the investigation of other drugs to alleviate symptoms or reverse the blistering process.

Prevention of trauma and blister management are primary goals in EB care. Therefore, it is recommended to use atraumatic dressings and bandages. Additionally, precautions should be taken regarding the selection of clothing, diapers, or underwear [59]. Bandages play a crucial role in dressing these patients and securing dressings, providing protection to areas that are more prone to shock or chafing [60].

Currently, advanced therapies are being adopted to improve healing with the goal of correcting the underlying genetic pathology or mitigating its effects. These therapies ultimately aim to correct the absence or reduction in anchoring proteins located at the dermal-epidermal junction. These encompass protein therapies, cell therapies, and gene therapies [61], and, together with the use of non-adherent traumatic dressings or silicones, form the therapeutic arsenal required to address EB [37,62,63,64,65,66].

Protein therapies involve obtaining the deficient protein through recombinant methods and subsequently applying it. This approach has been used for the restoration of collagen VII in Dystrophic EB (RDEB) and laminin 332 in Junctional EB [65].

On the other hand, cell-based therapies involve the local or systemic application of cells that will produce the deficient binding proteins or differentiate into other cell lines to achieve this [61,65]. Cell therapy encompasses a variety of therapies including primary keratinocytes, fibroblasts, hematopoietic cells, and mesenchymal / stromal stem cells [11].

Our work includes two trials that evaluated the use of fibroblasts [51,56]. In addition to keratinocytes, fibroblasts are the main source of type VII(C7) collagen formation in the skin. Animal studies have been performed in which allogeneic fibroblasts were administered to intact skin of mice with recessive dystrophic Recessive Epidermolysis Bullosa (RDEB). C7 re-expression was observed at a high dose of 5 × 10^6^ cells/cm^2^ [67], but not at a low dose of 1 × 10^6^ cells/cm^2^ [68]. Based on this high dose, Wong et al. reported clinical evidence of intradermal injection of fibroblasts in five patients with EBDR [69]. Allogeneic and haploidentical (from one parent) fibroblasts increased C7 expression for approximately 3 to 9 months. In other clinical trials, allogeneic fibroblasts were injected into the base or margin of chronic wounds with EBDR at a dose of 2.5 to 5 × 10^6^ cells/cm^2^, which promoted significant wound healing and, in some individuals, increased C7 expression for 3 to 12 months [51,56,69]. Since injected fibroblasts were not detectable after 2 weeks, therapeutic effects were mainly attributed to increased expression of the endogenous COL7A1 mutant [70]. However, the painful intradermal injection process must be taken into account. This was intolerable for many patients [68,70,71].

Finally, gene therapies have concentrated on replacing genes in recessive forms of EB and silencing genes in dominant forms, thus correcting the diseased genotype that causes the EB phenotype. A notable clinical trial in gene therapies is B-VEC, which is an investigational topical therapy aimed at restoring the C7 protein through the administration of COL7A1 (the gene that encodes the aforementioned protein) [65].

Regarding B-VEC therapy, our review includes two RCTs for wound care [42,43]. On the one hand, in terms of complete wound healing and closure, there is a significant difference compared with placebo treatment in both studies. In Guide, et al. [42] at 6 months (*p* = 0.002) and 3 months (*p* < 0.001) and in Gurevich I. et al. [43], *p* = 0.0026. On the other hand, 50% of B-VEC wounds and 7% of placebo wounds were closed at 3 and 6 months in the study carried out by Guide et al. [42], a very positive result. It should be mentioned that, in the article by Gurevich et al. [43] the wounds in the placebo group showed fluctuations in the wound closure rate.

Compared with other reviews, Prodinger et al., in 2019, showed that current therapies focus only on the treatment of wounds and pain. Therefore, the authors state that new therapeutic approaches are urgently needed. Therefore, a review was carried out in which gene-, protein- and cell-based therapies are mentioned. In addition, they refer to two treatments included in this review, the use of topical calcipotril (enhances skin immunity and tissue repair through hCAP18 potentiation) and diacerin (negatively regulating interleukin-1beta (IL-1ß) activity which is related to increased blistering in some EB). Blistering was found to be related to matrix metalloproteinase (MMP)-9 and the chemokine CXCL8/IL8, whose expression is dependent on the IL-1β signalling pathway. This product modulates the tissue microenvironment and reduces autoinflammatory effects in the skin of patients with EBS as potential agents to improve wound healing of the skin in patients with EBS [2].

A year later, in 2020, Has et al. referred to innovative therapies. He adds that research has been directed at palliating DEB and JEB, as opposed to EBS and EB despite their higher prevalence. He mentions replacement therapy for deficient genes and proteins as the only way to “cure” EB and that the administration of adjuvant therapies may help decrease the severity of the disease [72].

Continuing with the analysis of innovative therapies, two important therapies, considered orphan drugs for EB (“Orphan drug” is a drug used in rare diseases) are Oleogel S10, also called betulin or birch bark extract, which is known to be topically useful in the healing of EB wounds. The dry extract of birch bark promotes keratinocyte differentiation in vitro and in vivo [72,73]. Allantoin is a substance that has a keratolytic, bactericidal, anti-inflammatory and fibroblast proliferation and synthesis of extracellular matrix [74].

There is evidence on the use of oleogels from a previous review by. Schwieger-Briel A et al. conclude that oleogel-S10 accelerates wound reepithelialisation, and this is thought to be due to a bimodal pro-inflammatory and anti-inflammatory effect, as well as enhanced keratinocyte migration and differentiation. In particular, the active ingredient in Oleogel-S10 transiently up-regulates pro-inflammatory cytokines such as IL-6, IL-8 and TNF-α. Compared with our results, no substantial improvement in Ologel-S10 was observed with respect to pruritus, although in an RCT an improvement was found at 90 days with respect to pain [46,47,52].

Regarding allantoin use, studies show no improvement in healing time, pain management, and pruritus. It appears that the higher the concentration of the product, the better the results [48,50,55].

In evaluating other miscellaneous treatments, studies have been conducted with topical sirolimus and henna. Sirolimus attempts to inhibit a metabolic pathway, thus decreasing the translation of defective keratin proteins. Only one RCT was found that did not report statistically significant results compared with the control group. Henna (Lawsonia inermis) is a medicinal plant that has healing properties for wounds and their symptomatology. The application of henna has been shown to promote wound healing and alleviate itching. Furthermore, additional research has uncovered the antimicrobial and antifungal properties of henna, attributing these effects to the high concentrations of various components, including carbohydrates, anthraquinones, naphthoquinone derivatives, flavonoids, and phenolic compounds found in this plant. The only study on henna included in this review assesses the satisfaction of the patient and the physician. They reported high levels of satisfaction and a significant improvement in itching, burning, local heat, and redness of the area (*p* < 0.05), also in local pain, but not clinically significant [49,54].

We continue with two therapies used for local immune modulation, 1% diacerein and calcipotriol. Diacerein is a prodrug that inhibits the IL-1 converting enzyme. It is used for the systemic treatment of osteoarthritis and, topically, greatly reduces blistering. Calcipotriol, an active vitamin D3 analogue, has immunomodulatory properties, reduces pruritus, and improves wound healing [2,57].

In relation to calcipotriol, Prodinger et al. stated that all existing methods for the treatment of bacteria-infected wounds have some disadvantages. On the one hand, antiseptic baths, which are recommended for patients with dystrophic EB, are often time-consuming, exhausting, and painful for patients, as all dressings must be carefully removed. For this reason, the patient’s own immunity must be boosted. Antimicrobial peptides (AMPs) play a key role in this. These are part of the innate immune response of the body, serve as potent antibacterial substances that control pathogenic infections, and activate the innate immune system [75].

Following their argument, cathelicidin (hCAP18) is a prominent AMP in human epithelial cells, which serves to enhance host defences and appears to play a role in tissue repair and wound closure. In response to skin infections, hCAP18 is positively regulated on the skin and exhibits direct antimicrobial, antiviral, and antifungal activity [76,77].

Cathelicidin is directly upregulated by vitamin D [78]. Sun exposure, which leads to the production of the prehormone in the skin, serves as a distinct source of vitamin D. In patients with EB, limited exposure to the sun due to wound dressings and reduced outdoor activity can lead to vitamin D deficiency [79], resulting in reduced cathelicidin production and reduced antimicrobial defence [2].

Our meta-analysis includes the Guttmann-Grub et al. study which supports that topical treatment with low-dose calcipotriol resulted in a significant reduction in wound area on day 14 compared with placebo (88.4% vs. 65.5%, *p* < 0.05). Patients also reported a significant reduction in itching with calcipotriol ointment compared with placebo throughout treatment, as evidenced by itch scores of 3.16 vs. 4.83 (*p* < 0.05) and 1.83 vs. 5.52 (*p* < 0.05) on days 14 and 28, respectively [5].

Regarding diacerein, Prodinger et al. reported in their review that topical diacerein regulates IL-1 activity and reduces auto-inflammatory effects on the skin of patients with Epidermolysis Bullosa Simplex (EBS). The study demonstrated a reduction in the number of blisters by more than 40% in selected areas, and this effect remained significant during follow-up. Particularly, changes in the absolute number of blisters were significant only in the diacerein group, and no adverse effects were observed [57,58]. However, Prodinger et al. acknowledged the limitations of their study, including the small number of patients and the lack of invasive data acquisition due to the high clinical prevalence in children.

Our review includes a new study in addition to those reported by Prodinger et al. In 2023, Teng et al. evaluated the efficacy and safety of 1% diacerein ointment in the treatment of EBS. They considered a reduction of at least 60% in the body surface area of the EBS and a reduction of 2 points in the global assessment of the investigated person as a success criterion. The authors concluded that there were no significant differences between the groups and more studies are needed. However, all studies, including the new one, reported a decrease in blister size [54].

Therefore, our results partially support the findings of Prodinger et al. However, observing the results of our meta-analysis, significant progress is needed in both advanced therapies and conventional approaches to wound treatment, as well as in the treatment of pain and pruritus. This need is further underscored by the findings of Tang’s 2021 review, which included 65 studies and revealed a substantial prevalence of wounds among patients with EB. Sixty percent of the patients reported wounds covering more than 30% of their body. The study demonstrated an association between increased pain and pruritus with larger wounds. Chronic wounds have been shown to be larger and more painful than recurrent wounds [80].

Furthermore, Choi et al. evaluated the most recurrent pathological-related side effects, with 32 patients reporting mainly three effects: skin lesions and blisters (7/32 [23%]), itching (5/32 [16%]), and pain [81]. These findings align with those of Eng et al., who observed that the majority of the patients experienced itching (72/83, 85%), and the presence of itching did not vary according to the severity of skin disease reported by the patient [82].

Finally, atraumatic dressings are being studied to reduce the bacterial load, as well as to respect the wound bed. Carboxymethylcellulose dressings, which is a hydrocolloid that provides a moist environment that optimises healing, or biocellulose dressings, which favour water retention and adaptation of the wound, while allowing inspection of the lesion. Similarly, type I collagen dressings decrease the activity of collagenase and metalloproteinases, thus improving healing and keratinocyte migration. Finally, we found studies on cotton acetate dressings coated with dialkylcarbamoyl chloride. Dialkylcarbamoyl chloride has a microorganism trapping action to kill bacteria and fungi due to its hydrophobic capacity [37,38,39,40].

This study is not without limitations. First, the variability in treatments complicates the differentiation of their individual benefits, making it difficult to identify the most promising treatment. Furthermore, RCTs involve populations of different age groups, encompassing various types of EB and even introducing multiple subtypes of this diseasewithin the same study. Furthermore, the lesions examined exhibit different severities and locations. Control treatments also vary slightly among the reviewed studies, although most employ a placebo treatment with saline or a gel lacking an active ingredient. Secondly, most studies lack follow-up measurements to assess the long-term effects of the applied treatments.

## 5. Conclusions

Research on epidermolysis bullosa has advanced significantly in the last decade, resulting in improvements in wound and pain management. However, further progress is needed to reduce the symptoms of these patients. More clinical trials are needed to identify treatments that can reduce or prevent the onset of symptoms that significantly impact the quality of life of patients with epidermolysis bullosa.

Advanced therapies present a considerable challenge and offer added advantages compared with conventional treatments. On the one hand, dressings are being refined to allow bloodless removal as well as to minimise infection (e.g., dialkylcarbamoyl chloride (DACC) dressings). On the other hand, knowledge at the genetic level of the different subtypes of epidermolysis bullosa allows a targeted approach to specific needs, making it possible to treat deficiencies at the dermal level by topical interventions at different levels (protein, cellular and genetic). Therefore, the use of Oleogel S-101, V-BEC, allantoin and diacerein 1%, followed by the use of fibroblasts, has promising results.

Finally, it is recommended to continue conducting studies, based on existing research, to facilitate future comparisons and define a therapeutic arsenal for the management of the disease. In the same way, the type of epidermolysis bullosa that the patient has should be taken into account since many studies focus on DEB and JEB, with EBS being more prevalent. More studies are needed to effectively treat blistering, itching, and healing time in people with EB.

As a prospective line, it is recommended to increase the body of knowledge of cellular, protein and gene therapies, using methodologies identical to those published by previous authors. This will make it possible to use similar scales and thus be able to determine the real potential of each treatment. Also, adjuvant treatments could be carried out between genetic, protein or cellular therapies with the topical use of henna, calcipotriol, among others. This will make it possible to assess the synergies between different treatments.

## Figures and Tables

**Figure 1 healthcare-12-00261-f001:**
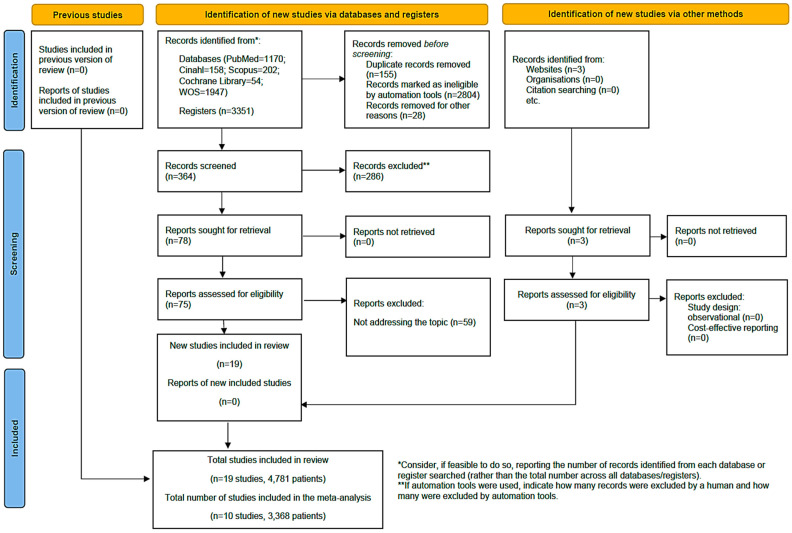
Flow diagram of the selection of articles [36].

**Figure 2 healthcare-12-00261-f002:**
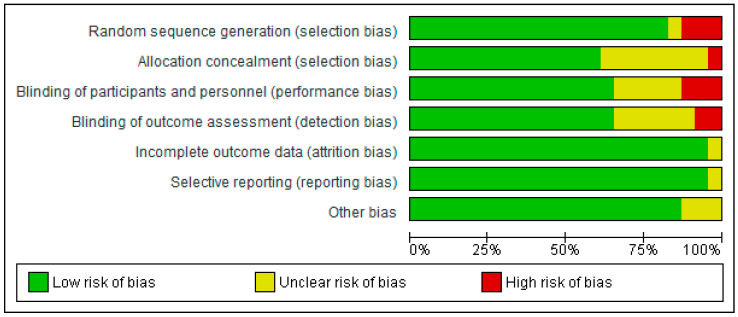
Bias assessment plots of all included studies.

**Figure 3 healthcare-12-00261-f003:**
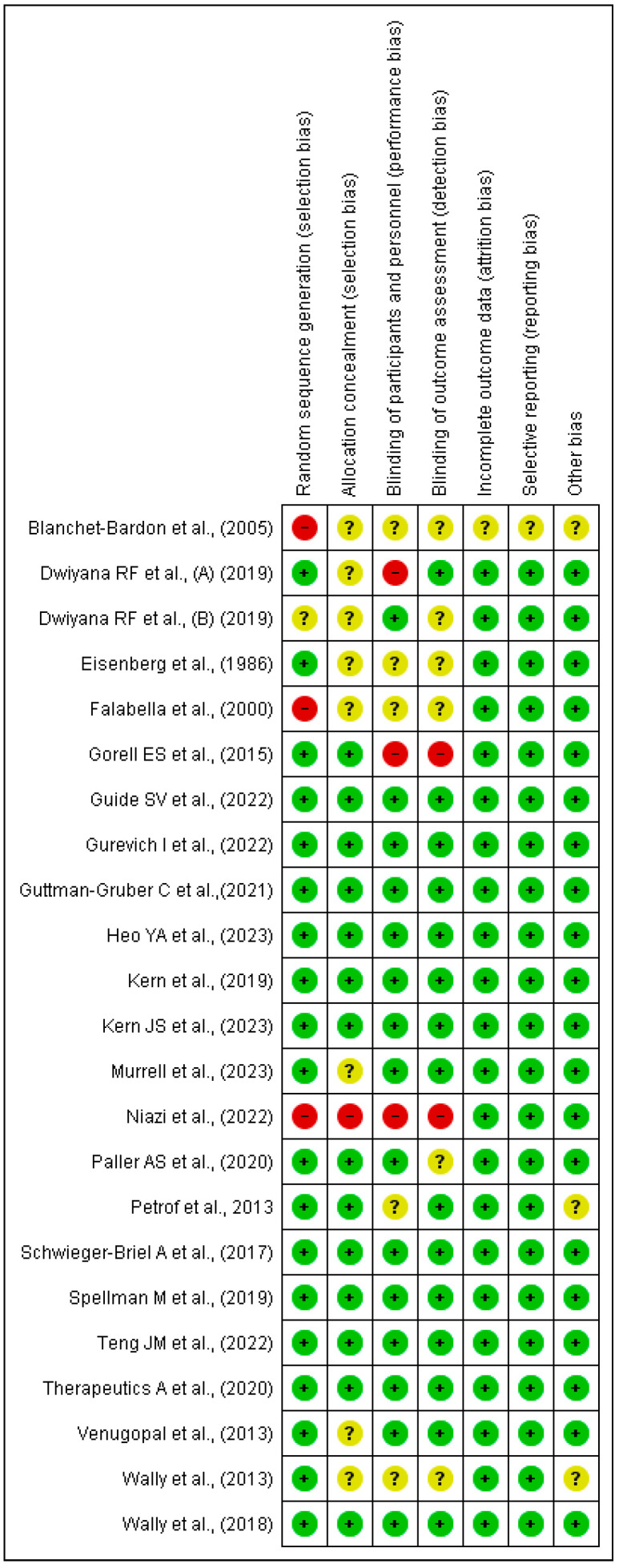
Bias assessment by a one-to-one summary plot [37,38,39,40,41,42,43,44,45,46,47,48,49,50,51,52,53,54,55,56,57,58].

**Figure 4 healthcare-12-00261-f004:**
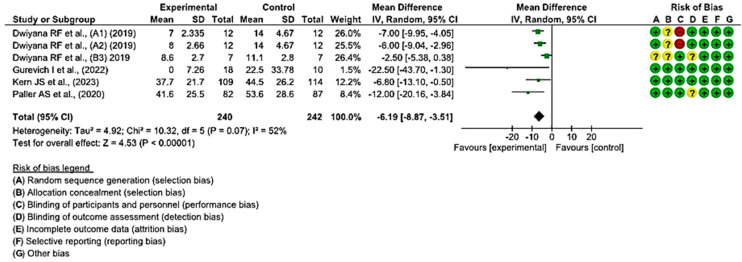
Efficacy of topical treatments in the time of complete wound healing [38,39,44,47,50].

**Figure 5 healthcare-12-00261-f005:**
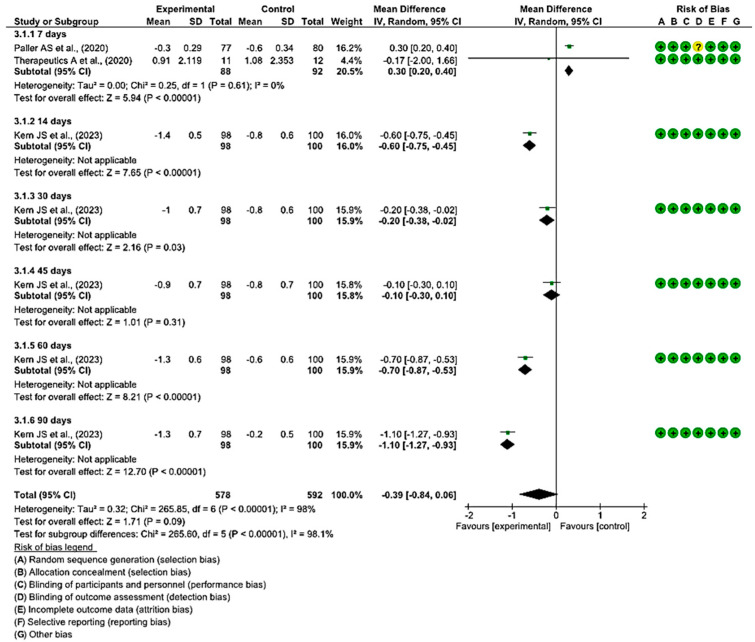
Efficacy of topical treatments on the change in WBF score for pain assessment [47,50,55].

**Figure 6 healthcare-12-00261-f006:**
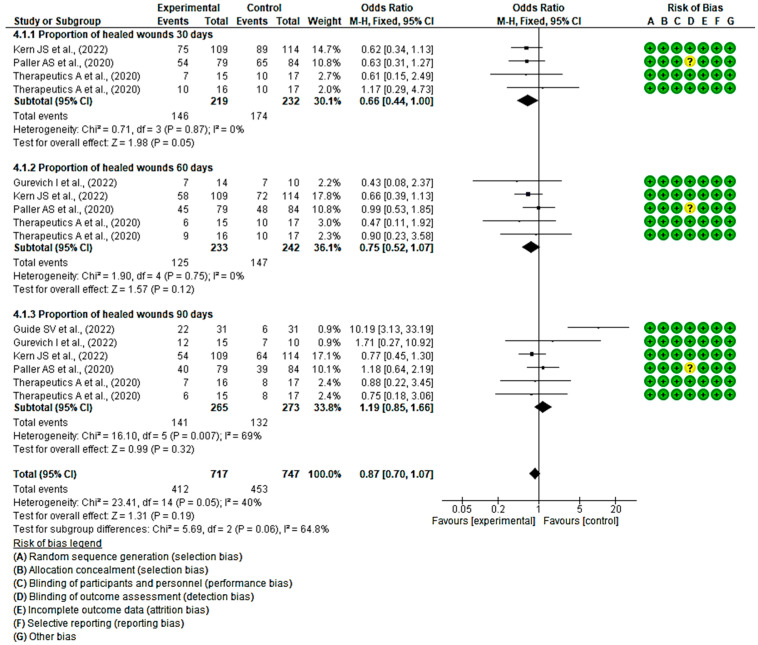
Efficacy of topical treatments on the proportion of wound healing within a specified timeframe [43,44,46,50,55].

**Figure 7 healthcare-12-00261-f007:**
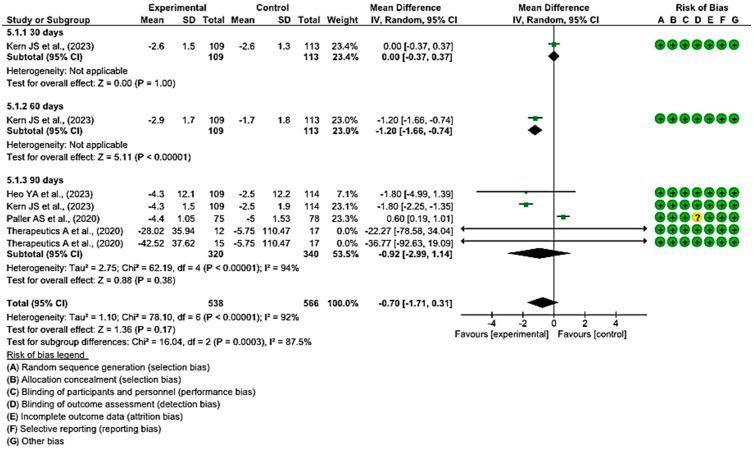
Efficacy of topical treatments in changes in BSA scores [5,47,50,54].

**Figure 8 healthcare-12-00261-f008:**
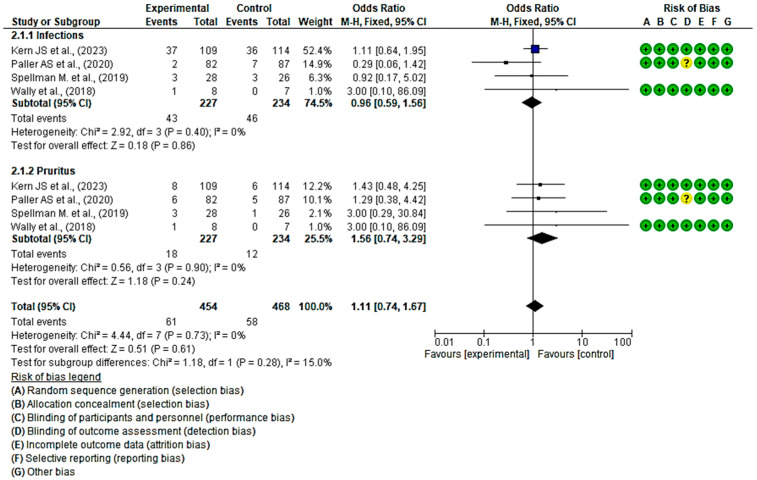
Efficacy of topical treatments in infections and pruritus [47,50,53,58].

**Table 1 healthcare-12-00261-t001:** Characteristics of the included studies.

Author, Country (Year). Evidence	Design. EB Type	Objective	Study Treatments	Results
Blanchet-Bardon, C. [37] France (2005) (1+)	Open, prospective, non-randomized, single-centre clinical trial. EBS, EB and DEB	To evaluate the acceptability, tolerance and efficacy of Urgotul wound dressing in the treatment of EB skin lesions.	10 × 10 cm Urgutol dressings.	High acceptability and effectiveness of Urgotul in the treatment of skin lesions in patients with EB.
Dwiyana RF et al. (1) [38] Indonesia (2019) (1+)	Single-blind, randomised, and controlled clinical trial. EBS and DEB	To compare the effectiveness of the three dressings in wounds of patients with BE by measuring complete healing times and the percentage of wound closure measured every three days.	Group 1: Biocellulose dressings Group 2: Carboxymethylcellulose dressings Group 3: Saline dressing (control)	There is a significant difference between group 1 and group 2, respectively, with group 3 (*p* < 0.05) with respect healing times. However, between groups 1 and 2 there are no significant differences (*p* > 0.05).
Dwiyana RF et al. (2) [39] Indonesia (2019) (1+)	Single-blind, controlled clinical trial. DEB and EBS	Compare the therapeutic efficacy of the two groups through the average time of wound closure and elimination of bacterial infection through clinical manifestations.	Group 1: DACC-coated cotton acetate dressing Group 2: A combination of normal saline dressing and 2% mupirocin ointment	Group 1 demonstrated faster effects than group 2 (*p* < 0.05) in wound closure. Both groups achieved clinical improvement on day 6, with no appreciable statistical differences (*p* = 1000).
Eisenberg, M [40] Australia (1986) (1+)	Controlled clinical trial. EB and RDEB.	Investigate the effect of occlusive and non-occlusive dressings on epidermal rejuvenation and dermal fibrosis of wounds in a group of children with RBD.	Group 1: HCD, occlusive, opaque, tan, 1.5 mm adhesive Group 2: Perforated, non-adhesive, oxygen-permeable plastic film covered by an absorbent layer (TELFA, Kendall Co, Boston). Group 3: PG, covered by an absorbent dressing.	The HCD dressing adhered easily to the normal surrounding skin and within 24 h the portion of the dressing directly over the wound was darker and softer than the adjacent portions.
Falabella, AF [41] Sweden (2000) (1+)	Open, uncontrolled study. EB.	To determine the safety and clinical effects of tissue-modified skin (Apligraf; Organogenesis Inc, Canton, Mass) on wound healing in patients with different types of EB.	Each patient received tissue-engineered skin on up to 2 wounds at each of 3 clinical visits: day 1, week 6, and week 12.	The tissue-engineered skin induced very rapid healing, was not clinically rejected, and was free of adverse effects. Patients and their families considered it to be more effective than conventional dressings for EB wounds.
Gorell ES., et al. [42] California. (2015) (1−)	Randomised, randomised, and unblinded clinical trial. EB and RDEB.	To compare the effectiveness of the three dressings in wounds of patients with BE by measuring complete healing times and the percentage of wound closure measured every three months. days.	Group 1: Dressing containing collagen (Helicoll) Group 2: Standard care dressings (Mepilex, mepitel, or Vaseline gauze)	The percentage of improvement in wounds treated with collagen-containing dressings was statistically significant (*p* = 0.03), as opposed to standard dressings (*p* > 0.99). In the control wounds, there were no changes in these parameters.
Guide SV., et al. [43] US (2022) (1++)	Phase 3, double-blind, randomized intrapatient, controlled trial. DEB y RDEB.	Assess the application of B-VEC in the wounds of patients with dystrophic epidermolysis bullosa	Group 1: B-VEC Group 2: Placebo	At 6 months, complete wound healing occurred in 67% of wounds exposed to B-VEC compared with 22% of those exposed to placebo. The complete cure of wounds at 3 months occurred in 71% of wounds exposed to B-VEC compared with 20% of those exposed to placebo. The mean change from baseline to week 22 in pain intensity during dressing changes was −0.88 with B-VEC and −0.71 with placebo.
Gurevich, I., et al. [44] US (2022) (1++)	Phase 1 and 2 trial, randomized and controlled. RDEB.	To evaluate B-VEC, an engineered, non-replicating COL7A1 containing a HSV-1 vector, for treating skin with EBDR. B-VEC restored C7 expression in keratinocytes, fibroblasts, mouse RDEB and human RDEB xenografts	Group 1: Patients with EBDR who received topical B-VEC Group 2: EBDR patients who received placebo Time: repeatedly for 12 weeks.	Cultures of keratinocytes and fibroblasts demonstrated C7 expression 48 h after B-VEC treatment. Dose-dependent increases in transduction efficiency were also demonstrated, targeting up to 100% of cells at a multiplicity of infection (MOI) of 1, 3 and 10, with a slowing of the proliferation at an MOI of 10.
Guttmann-Gruber C et al. [45] Austria (2021) (1++)	Monocentric phase II, crossover, randomized, double-blind and controlled clinical trial. DEB.	To evaluate the effectiveness of daily topical application of 0.05 µg/g calcipotriol ointment in reducing the size of the wound within a 4-week treatment regimen	Group 1: 0.05 µg/g calcipotriol ointment Group 2: Placebo	In wound size, on day 14, a reduction of 88.4% was observed in the calcipotriol treatment as opposed to 65.6% in the placebo treatment (*p* = 0.006). The fraction of completely closed wounds were higher in the treatment with calcipotriol but there were no statistically significant results (*p* = 0.413).
Heo, Y [5] Nueva Zelanda (2023) (1++)	EASE Phase III Pivotal, Double-Blind, Randomized, Controlled Trial. EB.	To analyse the use of topical birch bark extract gel (Oleogel-S10) in junctional EB in patients ≥6 months of age.	Group1: Birch bark extract (Oleogel-S10) Group 2: Control gel	Oleogel-S10 relative to control gel significantly increased the proportion of patients with first complete closure of the target wound within 45 days.
Kern, J. et al. [46] Australia, Germany, Switzerland, Ireland, United Kingdom. (2019) (1++)	EASE: Two-phase phase 3 study comprising a 90-day, double-blind, randomized, placebo-controlled phase. EB.	Support the determination of the efficacy and safety of Oleogel-S10 in EB	Group 1: with Oleogel-S10. Group 2: with placebo (based on the vehicle Oleogel-S10, formulated to have a consistency indistinguishable from that of the active product)	There are no results, according to the article it was expected that they would be in the second half of 2019
Kern, J et al. [47] Australia, Israel, Argentina, Germany, France, UK (2023) (1++)	EASE phase III, double-blind, randomized, controlled study. EB, DEB, JEB and EB Kindler.	To determine the efficacy and safety of topical gel Oleogel-S10 in EB.	Group 1: 109 patients treated with Oleogel-S10 Group 2: 114 with control gel	109 treated with Oleogel-S10, 114 with control gel Oleogel-S10 resulted in 41.3% of patients with first complete closure of the target wound within 45 days, compared with 28.9% in the control gel group (relative risk 1.44, 95% CI 1.01–2.05; *p* = 0.013).
Murell, A. et al. [48] Chicago. (2023) (1++)	Phase 3, multicentre, randomized, double-blind, vehicle-controlled study. EB, RDEB and JEB.	To evaluate the efficacy and safety of SD-101 cream at 6% versus vehicle (0% allantoin) in lesions in patients with BE.	Group 1: Randomly assigned to SD-101 at 6% (n = 82) Group 2: vehicle cream (n = 87)	There were no statistically significant differences between treatment groups in time to wound closure (HR, 1.004; 95% CI, 0.651, 1.549; *p* = 0.985) or proportion of patients with complete closure of the target wound within 3 months (OR 95% CI, 0.733 [0.365, 1.474]; nominal *p* = 0.390). Closure was observed with SD-101 6% versus vehicle in patients aged 2 to <12 years and those with total body treatment.
Niazi M et al. [49] Iran (2022) (1−)	Single-arm pilot clinical trial. RDEB.	Efficacy of the use of topical henna on wounds for their management and for the improvement of itching in patients with EB	Evolution of topical henna application in 4 weeks.	A significant improvement was observed (*p* > 0.05). 5/7 patients reported an improvement in henna treatment for pruritus, preferring it over other treatments.
Paller AS et al. [50] USA (2020) (1++)	Multicentre, double-blind, controlled clinical trial. EBS, RDEB, and intermediate JEB.	To assess the time to complete target wound closure within 3 months, the proportion of patients with target wound closure within months 1, 2, and 3, change in BSA index and change in pain	Group 1: SD 101 (6% allantoin) Group 2: SD-101 (0% allantoin	Group 1 wounds closed faster in short periods of time (at least 1 month), these also had >OR = 5% BSA
Petrof, G. et al. [51] Manchester (2013) (1++)	Prospective, double-blind, randomized, vehicle-controlled phase II trial. EB and RDEB.	To assess the effects of injecting of allogeneic fibroblasts into the margins of chronic erosions in individuals with RDEB	Grupo 1: fibroblasts Grupo 2: vehículo alone	Treatment difference between fibroblasts and vehicles was 235% (95%CI 35 to 435, *p* = 0025) at day 7, 1915% (95%CI 336 to 4166, *p* = 0089) at day 14 and −2883% (95%CI 797 to 6563, *p* = 011) at day 28.
Schwieger-Briel A et al. [52] Germany and Switzerland (2017) (1++)	Prospective, Controlled, Blinded, and Open-Label Phase II Pilot Trial. RDEB y DDEB.	The healing of wounds treated with and without topical Oleogel-S10 was compared by the speed of re-epithelialization and the percentage of wound epithelialization.	Group 1: Oleogel-S10 + non-adhesive dressing. Group 2: Non-adhesive dressing alone.	In 42% of the cases the epithelialization of the wounds of the intervention group is superior. Oleogel group was considered the same as control.
Spellman M et al. [53] USA (2019) (1++)	International, multicentre, randomized, double-blind, vehicle-controlled, parallel group clinical trial. EBS.	To compare the effectiveness of 1% diacerein ointment with vehicle ointment based on reduction in BSA of the EBS lesions being treated when applied once daily for 8 weeks.	Group 1: Cream diacerein 1% Group 2: Placebo	The proportion of subjects who achieved a reduction of ≥60% BSA of EBS lesions within the evaluation area from baseline to week 8 was 57.1% of participants in the diacerein group and 53.8% in the placebo group.
Teng JM et al. [54] USA (2022) (1++)	Prospective, double-blind, randomized, placebo-controlled crossover clinical trial. EBS	To assess EBDASI index from week 0 to week 12 of treatment and variations in the 5D pruritus scale in weeks 0–12.	Group 1: Sirolimus 2% Group 2: Placebo	The EBDASI index went from 2.6 at week 0 to 2.9 at week 12 in the sirolimus treatment group and from 3.5 to 2.5 in the placebo group. The 5D pruritus scale varies in group 1 from 12.8 to 12.5 in weeks 0 to 12 and, in group 2, from 11.5 to 11.8.
Therapeutics A., et al. [55] US (2020) (1++)	Triple-blind, randomized, parallel assignment clinical trial. EBS, RDEB and JBE no Herlitz.	To evaluate the participants with complete closure of the target wound within one, two or three months after the start of treatment.	Group 1: Cream dermal SD-101 (6% allantoin) Group 2: Cream dermal SD-101 (3% allantoin) Group 3: Vehicle (0% allantoin)	Complete wound closure is achieved within 3 months after treatment in 60% of group 1, 56.3% of group 2 and 52.9% of group 3. The change in BSAI is −28.02 in group 1, −42.52 in group 2 and −5.75 in group 3. Pain in group 1 it was 0.91, group 2 −0.7 and group 3, 1.08
Venugopal, SS. et al. [56] Sydney, Perth, Australia, California. Japan (2014) (1++)	This was a phase II double-blinded randomized controlled trial of intralesional allogeneic. EB and RDEB.	To Study the Application of Intralesional Cultured Allogeneic Fibroblasts in Suspension Solution Versus Suspension Solution Alone for Wound Healing in RDEB-GS	Group 1: Fibroblasts were transported in suspension solution (Plasma-Lyte148 [Baxter, Boronia, VIC] with 2% Albumex20 [CSL Bioplasma, Parkville, VIC]) at 20 3 106 cells/mL and viability confirmed on receipt (99%). Group 2: Suspension solution alone was transported for placebo injections.	All wounds healed significantly more rapidly with fibroblasts and vehicle injections, with an area decrease of 50% by 12 weeks, compared with noninjected wounds. Collagen VII expression increased to a similar degree in both study arms in wounds from 3 of 5 patients.
Wally, V. et al. [57] Austria (2013) (1+)	Pilot study, open and withdrawal phase, controlled, randomized and double-blind.	To assess if Diacerin is capable of reducing the formation of blisters.	Open phase of six weeks with application of 1% diacerin under the armpits to all patients Second phase. Randomized and placebo-controlled	Statistically significant reduction in blisters within the first two weeks of Phase 1, which remained stable until the end of the study. At Phase 2 no loss of efficacy could be observed, and the primary endpoint was omitted.
Wally V et al. [58] Austria, France and the Czech Republic (2018) (1++)	Double-blind, randomized, placebo-controlled phase 2/3 clinical trial. EB.	Proportion of patients with a recurrence of the initial number of blisters by 10% at the end of treatment Proportion of patients with a reduction in the number of blisters of more than 40% in the BSA from the beginning to the end of treatment Evaluation of pruritus and pain using the VAS scale.	Group 1: Cream diacerein 1% Group 2: Placebo	The streaks are reduced by more than 40% in 86% of patients treated with diacerein and 14% of the placebo group in the first treatment episode (4 weeks). At the end of follow-up, all patients in group 1 show a reduction of more than 40% of blisters, compared with 57% of placebo patients.

B-VEC: Beremagene Geperpavec; BSA: Body Surface Area; BSAI: Body Surface Area Index; CI: Confident interval; DACC: Dialkylcarbamoylchloride; EB: Epidermolysis Bullosa; HCD: Hydrocolloid oxygen-impermeable dressing; HR: Hazard Ratio; HSV-1: Herpes Simplex Virus Type 1; IGA: Investigator’s Global Assessment; OR: Odds Ratio; PG: Paraffin gauze; VAS: Visual Analogue Scale.

**Table 2 healthcare-12-00261-t002:** Degrees of recommendations.

Assessment of Certainty							№ of Patients		Effect		Certainly	Outcomes
№ of Studies	Study Design	Risk of Bias	Inconsistency	Indirect Evidence	Imprecision	Other Considerations	Topical Treatments	Placebo	Relative (95% CI)	Absolut (95% CI)		
5	RCTs	Serious	It is not serious	Serious	Serious	Publication bias is strongly suspected	240	242	-	MD −5.59 (−7.2 to −3.97)	⨁◯◯◯ Very low	Time of complete wound healing (days)
4	RCTs	Serious	Very serious	Serious	Serious	Publication bias is strongly suspected	43/227 (18.9%)	46/234 (19.7%)	OR 0.96 (0.59 to 1.56)	6 minus per 1000 (−70 to 80)	⨁◯◯◯ Very low	Side effects—Infections
4	RCTs	Serious	Very serious	It is not serious	It is not serious	Publication bias is strongly suspected	18/227 (7.9%)	12/234 (5.1%)	OR 1.56 (0.74 to 3.29)	26 plus per 1000 (−13 to 100)	⨁◯◯◯ Very low	Side effects—Pruritus
2	RCTs	Very serious	Very serious	Serious	Serious	Publication bias is strongly suspected	88	92	-	MD 0.3 (0.2 to 0.4)	⨁◯◯◯ Very low	Change in WBF scores—7 days
1	RCTs	Very serious	Very serious	Serious	Serious	Publication bias is strongly suspected	98	100	-	MD −0.6 (−0.75 to −0.45)	⨁◯◯◯ Very low	Change in WBF scores—14 days
1	RCTs	Very serious	Very serious	Serious	Serious	Publication bias is strongly suspected	98	100	-	MD −0.2 (−0.38 to −0.02)	⨁◯◯◯ Very low	Change in WBF scores—30 days
1	RCTs	Very serious	Very serious	Serious	Serious	Publication bias is strongly suspected	98	100	-	MD −0.1 (−0.3 to 0.1)	⨁◯◯◯ Very low	Change in WBF scores—45 days
1	RCTs	Very serious	Very serious	Serious	Serious	Publication bias is strongly suspected	98	100	-	MD −0.7 (−0.87 to −0.53)	⨁◯◯◯ Very low	Change in WBF scores—60 days
1	RCTs	Very serious	Very serious	Serious	Serious	Publication bias is strongly suspected	98	100	-	MD −1.1 (−1.27 to −0.93)	⨁◯◯◯ Very low	Change in WBF scores—90 days
3	RCTs	Serious	Very serious	It is not serious	It is not serious	Publication bias is strongly suspected	146/219 (66.7%)	174/232 (75.0%)	OR 0.66 (0.44 to 1.00)	86 minus per 1000 (−181 to 0)	⨁◯◯◯ Very low	Proportion of healed wounds 30 days
4	RCTs	Serious	Very serious	It is not serious	It is not serious	Publication bias is strongly suspected	125/233 (53.6%)	147/242 (60.7%)	OR 0.75 (0.52 to 1.07)	70 minus per 1000 (−162 to 16)	⨁◯◯◯ Very low	Proportion of healed wounds 60 days
5	RCTs	Serious	Very serious	It is not serious	It is not serious	Publication bias is strongly suspected	141/265 (53.2%)	132/273 (48.4%)	OR 1.19 (0.85 to 1.66)	43 plus per 1000 (−40 to 125)	⨁◯◯◯ Very low	Proportion of healed wounds 90 days
1	RCTs	Serious	Very serious	It is not serious	It is not serious	Publication bias is strongly suspected	109	113	-	MD 0 (−0.37 to 0.37)	⨁◯◯◯ Very low	Change in skin BSA index—30 days
1	RCTs	Serious	Very serious	It is not serious	It is not serious	Publication bias is strongly suspected	109	113	-	MD −1.2 (−1.66 to −0.74)	⨁◯◯◯ Very low	Change in skin BSA index—60 days
4	RCTs	Serious	Very serious	It is not serious	It is not serious	Publication bias is strongly suspected	320	340	-	MD −0.52 (−0.82 to −0.22)	⨁◯◯◯ Very low	Change in skin BSA index—90 days

CI: confidence interval; MD: Mean Difference; OR: Odds ratio; RCTs; Randomized Clinical Trial; ⨁⨁⨁◯ = level of recommendation.

## Data Availability

The data supporting this research can be requested by e-mailing the corresponding author.

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
