# Peer review of "Management of Skin Lesions in Patients with Epidermolysis Bullosa by Topical Treatment: Systematic Review and Meta-Analysis"

_healthcare, 2024, doi:10.3390/healthcare12020261_

Round 1
Reviewer 1 Report
Comments and Suggestions for Authors
The authors present a comprehensive review of managing skin defects in patients with Epidermolysis bullosa.
The Methodology is sound and follows recomendations for systemic reviews and meta-analysis.
I have however some minor comments:
1. You state that: 'A secondary objective was to analyse the effectiveness of diacerein, Oleogel S-10, allantoin, calcipotriol, B-VEC, dressings, sirolimus and henna...' Was this a specific aim or were those the topicals you found data on in the literature and then aimed to describe them? This is unclear. Also the sentence: 'there is a greater desire to investigate Oleogel-S10 (n=3), Diacerin (n=3) and allantoin (n=3)...' Greater than what? And what are the n referring to?
2. Could you consider a brief description of BSA and WBF scores. These are mentioned several times without explanation on scoring, meaning of scores, etc.
3. When describing toppicals and dressings could you consider giving better information on these. For instance: B-VEC, SD-101, DACC-coated cotton acetate dressing..' This is an abbreviation and not every reader can be expected to understand what this procedure is.
4. Other:
- Table 2 is very crowded with overlapping columns. Consider a landscape layout.
- Appendices: SE, MD, BSA, OR and any abbreviations should be explained in full, as well as uniots used or description of what the x and y axis actually tell the reader.
- You describe symptoms as a list including: ulcers, sores, erosions. What is the difference? Later you use wounds. Sores is not a medical term I think.
- Typos: Line 172 and 173: The sentence seems to end abruptly and the next is not part of it.
Comments on the Quality of English LanguageSome sentences have a structure that is strange and difficult to understand clearly: e.g.:
- Dermatologically, the management of painful lesions is predisposed to infections and slow healing.' The management is not predisposed to infections.
- OR: Caring for these patients requires specific attention to disease lesions, which present a challenge in daily clinical activities, without a consensus supported by scientific evidence in the treatment of local lesions...
-Starting a pararaph with 'This emphasises...' It is not very clear what is referred to.
Author Response
Dear Editors and Reviewers,
Thank you for reviewing the manuscript and for your comments. Please find below in the text the response to each comment. All the changes in the manuscript have been modified with the marked/tracked version.
The authors present a comprehensive review of managing skin defects in patients with Epidermolysis bullosa.
The Methodology is sound and follows recomendations for systemic reviews and meta-analysis.
I have however some minor comments:
- You state that: 'A secondary objective was to analyse the effectiveness of diacerein, Oleogel S-10, allantoin, calcipotriol, B-VEC, dressings, sirolimus and henna...' Was this a specific aim or were those the topicals you found data on in the literature and then aimed to describe them? This is unclear. Also the sentence: 'there is a greater desire to investigate Oleogel-S10 (n=3), Diacerin (n=3) and allantoin (n=3)...' Greater than what? And what are the n referring to?
Thank you very much for your comment. The objectives have been simplified in the summary. We believe that this makes them clearer for the reader. On the other hand, the n refers to the number of studies included in this review. After observing the results and future clinical trials, we observed a trend in certain products.
- Could you consider a brief description of BSA and WBF scores. These are mentioned several times without explanation on scoring, meaning of scores, etc.
Thank you very much for your comment. More information on both scales has been attached.
- When describing toppicals and dressings could you consider giving better information on these. For instance: B-VEC, SD-101, DACC-coated cotton acetate dressing..' This is an abbreviation and not every reader can be expected to understand what this procedure is.
Thank you very much for your comment. Abbreviations have been explained.
- Other:
- Table 2 is very crowded with overlapping columns. Consider a landscape layout.
Thank you very much for your comment. The table has been laid out in landscape.
- Appendices: SE, MD, BSA, OR and any abbreviations should be explained in full, as well as uniots used or description of what the x and y axis actually tell the reader.
Thank you very much for your comment. All abbreviations and units of measurement have been explained.
- You describe symptoms as a list including: ulcers, sores, erosions. What is the difference? Later you use wounds. Sores is not a medical term I think.
Thank you very much for your comment. The term sores has been deleted.
- Typos: Line 172 and 173: The sentence seems to end abruptly and the next is not part of it.
Thank you very much for your comment. The sentence has been completed
Some sentences have a structure that is strange and difficult to understand clearly: e.g.:
- Dermatologically, the management of painful lesions is predisposed to infections and slow healing.' The management is not predisposed to infections.
- OR: Caring for these patients requires specific attention to disease lesions, which present a challenge in daily clinical activities, without a consensus supported by scientific evidence in the treatment of local lesions...
-Starting a pararaph with 'This emphasises...' It is not very clear what is referred to.
All the sentences you comment on have been reviewed. Many thanks for your appreciation of the English revision. The manuscript was revised by a professional translator who is a native English speaker. However, it has been rechecked for flaws in the wording.
Thank you very much for your valuable comments that have allowed us to improve the manuscript.
Reviewer 2 Report
Comments and Suggestions for Authors
Line 44, Line 50, Line 52: avoid using several references for a statement like [1-4]. check other similar citation
Line 66-70: for this classification, it would be great if authors provide an image covering all types.
Line 92: "Therefore, this systematic review with meta-analysis aims to evaluate the effectiveness of local treatments on wounds in patients with EB", the main aim of the study seems to be similar to the previous works, making it more unique: (https://www.mdpi.com/2077-0383/12/3/1139)
Line 104: regarding the keywords, I am a bit curious about the final selected keywords. I believe that considering the main treatments for epidermolysis bullosa, there are other keywords that the authors did not consider. For instance, there are novel hydrogel-based strategies to treat epidermolysis bullosa. Tissue engineering, nanofiber-based, and nanoparticle-based studies have been also published. By considering these keywords and more related keywords the number of final selected papers will be increased.
The method has been described very well. I enjoyed it.
the results need a kind of explanation by which the authors explain how these results relate to the research hypotheses.
I studied the discussion and I found out that in some parts the authors did not pay attention to some biological points in the reviewed papers. I mean the authors mostly did a kind of comparison. for example Lines 398-400, and lines 423-429, please check other parts of the discussion and provide more biological information. You did this very well in some parts like Lines 443-446.
The conclusion was not written very well. please rewrite it and try to explain some main conclusions from your study based on your main aims of the study.
Besides, it would be great if you have some technical and method-based recommendations in the conclusion part as future perspectives.
Author Response
Dear Editors and Reviewers,
Thank you for reviewing the manuscript and for your comments. Please find below in the text the response to each comment. All the changes in the manuscript have been modified with the marked/tracked version.
- Line 66-70: for this classification, it would be great if authors provide an image covering all types.
Thank you very much for your comment. A figure has been attached as an appendix to the classification. Appendix 1.
2.Line 92: "Therefore, this systematic review with meta-analysis aims to evaluate the effectiveness of local treatments on wounds in patients with EB", the main aim of the study seems to be similar to the previous works, making it more unique: (https://www.mdpi.com/2077-0383/12/3/1139)
Thank you very much for your comment. Our work performs a statistical analysis through meta-analysis, providing a more objective view than a narrative review or a state of the art. In addition, it focuses on the approach to injuries.
3.Line 104: regarding the keywords, I am a bit curious about the final selected keywords. I believe that considering the main treatments for epidermolysis bullosa, there are other keywords that the authors did not consider. For instance, there are novel hydrogel-based strategies to treat epidermolysis bullosa. Tissue engineering, nanofiber-based, and nanoparticle-based studies have been also published. By considering these keywords and more related keywords the number of final selected papers will be increased.
Thank you very much for your comment. There are indeed studies incorporating topical therapies with the use of hydrogel, tissue engineering (we have incorporated only one study), use of nanofibres and nanoparticles. However, they did not meet our inclusion criteria. In order to perform the meta-analysis we needed the highest level of evidence, so they should be clinical trials that allow us to have an intervention versus control group. Databases and search engines did not identify clinical trials for these treatments.
4.The method has been described very well. I enjoyed it.
Thank you very much for your comment.
5.the results need a kind of explanation by which the authors explain how these results relate to the research hypotheses.
Thank you very much for your comment. This information has been included in the results section as you properly recommended.
6.I studied the discussion and I found out that in some parts the authors did not pay attention to some biological points in the reviewed papers. I mean the authors mostly did a kind of comparison. for example Lines 398-400, and lines 423-429, please check other parts of the discussion and provide more biological information. You did this very well in some parts like Lines 443-446.
Thank you very much for your comment. Further explanations of treatments at the biological and genetic level have been included.
7.The conclusion was not written very well. please rewrite it and try to explain some main conclusions from your study based on your main aims of the study.
Thank you very much for your comment. The conclusions have been rewritten based on the objectives.
- Besides, it would be great if you have some technical and method-based recommendations in the conclusion part as future perspectives.
Thank you very much for your comment. A clinical prospective study has been added.
Thank you very much for your valuable comments that have allowed us to improve the manuscript.
Round 2
Reviewer 2 Report
Comments and Suggestions for Authors
Thanks for addressing the comments.